# Synthesis of Zr-89-Labeled Folic Acid-Conjugated Silica (SiO_2_) Microwire as a Tumor Diagnostics Carrier for Positron Emission Tomography

**DOI:** 10.3390/ma14123226

**Published:** 2021-06-11

**Authors:** Gun Gyun Kim, Hye Min Jang, Sung Bum Park, Jae-Seon So, Sang Wook Kim

**Affiliations:** 1Department of Advanced Materials Chemistry, Dongguk University, Gyeongju 38066, Korea; gungyun88@dongguk.ac.kr (G.G.K.); jhmindy96@gmail.com (H.M.J.); 2Department of Safety Engineering, Dongguk University, Gyeongju 38066, Korea; parksungbum@dongguk.ac.kr; 3Department of Medical Biotechnology, Dongguk University, Gyeongju 38066, Korea

**Keywords:** silica microwires, PET imaging, Zirconium-89, tumor diagnostics, drug delivery system

## Abstract

This study evaluated the in vivo behavior and accumulation of silica particles in the form of wires, which were actively studied as drug carriers along with spheres, using positron emission tomography (PET). Wire-shaped silicon dioxide (SiO_2_) was synthesized at micro-size, using anodic aluminum oxide (AAO), a template, and folic acid (FA), which specifically binds folate receptors (FR) which are overexpressed in many cancers, and which was bound to the wire’s surface to confirm its possible use as a cancer diagnostic agent. In addition, for evaluation using PET, the positron-emitting nuclide ^89^Zr (t_1/2_ = 3.3 days) was directly bonded to the hydroxyl group (-OH) on the particle surface. The diameter and shape of the synthesized silica microwires (SMWs) were confirmed using SEM and TEM, the chemical bonding of FA was confirmed through FT–IR and NMR, and the labeling of ^89^Zr was measured by means of radio-thin-layer chromatography (TLC) measurement. Folic acid-conjugated SMWs (FA-SMWs) were found to have a low receptor-mediated uptake in cell internalization evaluation, but in PET studies, FA-SMWs stayed longer at the tumor site. In conclusion, we successfully synthesized a homogeneous silica microwire for drug delivery, we confirmed that the FA-conjugated sample remains at the tumor site for a relatively longer time, and we have reported the characteristic in vivo behavior of ^89^Zr-FA-SMWs.

## 1. Introduction

The field of micro/nanomaterial drug delivery research is growing rapidly and uses artificial synthetic materials of several micrometers and several nanometers in length [1]. This scale is similar to the size of proteins and biological polymers in the body [2,3]. This technology is used as a research tool in various fields as it can design complex characteristics, and it is being studied for its ability to overcome biological barriers, especially in the delivery of drugs and the development of various disease and disability treatments [4,5,6]. Physiologically active molecules, target sequences, fluorescent or radioactive isotopes, and biocompatible coatings can be introduced to improve the function of carriers [6,7,8]. Research on the size and shape of carriers contributes to the understanding of in vivo behavior and tumor accumulation [9,10,11]. Micro/nano-sized particles show a high accumulation in tumor tissues through slack capillary vessels close to the tumor and can be used to accrete concentrations of drugs in cancer cells, while avoiding toxicity to normal tissues by conjugating molecules with tumor affinity [12,13,14,15]. Furthermore, when nanoparticles are combined with certain receptors and then enter the cells, they are usually endocrine-resistant through receptor-mediated endocrine distribution, bypassing detection by the P-glycoprotein, one of the main drug-resistant mechanisms [16,17]. Despite offering many advantages as a drug carrier system, nanoparticles still have many limitations that need to be addressed, such as poor oral bioavailability, unstable circulation, insufficient tissue distribution, and toxicity [1,18,19,20]. Along with size, particle shapes can affect the behavior of circulating drug carriers as well as in cell–particle interactions. Particle size is reported to influence opsonization based on the relationship between particle size and curvature, which, in turn, is an important characteristic that influences almost all particle functions such as degradation, flow properties, clearance, and absorption mechanisms in the body [21]. In general, most micro-drug transporters reported in research and clinical applications are mostly spherical because the researcher adopted an ‘upward’ method of production. A typical upward method in the field of drug delivery systems is the sol-gel method. This method is achieved by hydrolysis and condensation reactions of inorganic materials to form a three-dimensional network structure, including the formation of an inorganic colloidal suspension (sol) and gelation of the sol in a continuous liquid phase (gel). Most synthetic particles form spherical or near-spherical shapes, but drug carriers using particles such as ellipsoids, rods, wires, and disks other than spheroids are also being studied and exhibit different behaviors in the body. Despite use of the sol-gel method, it is possible to synthesize wire-shaped particles by introducing a specific template (AAO) [22]. Recent non-invasive imaging techniques including positron emission tomography (PET) [23], have made great advances in dating and perform a consequential role in the personalized medical field, providing practical solutions to various incurable diseases such as cancer [24]. There is increasing attention being given to using newly rising radioactive isotopes (RIs) including ^89^Zr (t_1/2_ = 78.4 h) [25], ^64^Cu (t_1/2_ = 12.7 h) [26], ^86^Y (t_1/2_ = 14.7 h) [27], and ^68^Ga (t_1/2_ = 67.6 m) [28] for PET [6]. These RIs have been extensively studied, of which ^89^Zr is a promising RI for long-term tumor diagnostics study and in vivo distribution research, indicating the ease of relatively long-term production, favorable nuclear decay characteristics, and appropriate chemical binding properties [25,29,30]. In this work, we synthesized a folate-receptor-friendly drug carrier that combines folic acid with silica microwire. ^89^Zr was labeled on a folic acid-conjugated silica microwire for in vivo behavior evaluation using PET, and separately from the tail intravenous injection. We examined the behavior of the silica microwire injected on the tumor side to evaluate its in vivo behavior after its accumulation in the tumor.

## 2. Materials and Methods

### 2.1. Materials

A tetraethyl orthosilicate (TEOS), (3-aminopropyl) triethoxysilane (APTES), folic acid (FA), triethylamine (TEA), N-hydroxy succinimide (NHS), 1-ethyl-3-(3-dimaethylaminopropyl), carbodiimide (EDC), dimethylformamide (DMF), and anodic aluminum oxide (AAO) template were purchased from Sigma-Aldrich (St. Louis, MO, USA). Dulbecco’s modified Eagle’s medium (DMEM), phosphate buffered saline (PBS), HEPES buffer solution, human serum (HS), and fetal bovine serum (FBS) were purchased from Gibco BRL Life Technologies (Waltham, MA, USA). CT-26 and MDA-MB-231 cells were procured from the Korean cell line bank (Seoul, Korea). Balb/c mice (20 ± 1.5 g, female) were purchased from Orient Bio (Seongnam, Korea). ^89^Zr was procured from the Korea Atomic Energy Research Institute (KAERI, Jeongeup, Korea) and produced using RFT-30 (30 MeV cyclotron). Radioactivity was measured using a CRC-15R ionizing chamber (Capintec, Florham Park, NJ, USA). Radiochemical yield was assessed using an AR-2000 radio-TLC imaging scanner (Bioscan, Santa Barbara, CA, USA). The in vitro stability of the radiolabeled sample was measured with a Wizard-1470 automatic gamma counter (Perkin Elmer, Waltham, MA, USA). Small animal PET imaging was carried out using a Genesis 4 (Sofie Biosciences, Culver City, CA, USA) machine. Dynamic light scattering analysis was measured using a Zeta-sizer (Malvern, UK) and MN401 (Microtrac, York, PA, USA).

### 2.2. Synthesis of Silica Macro-Wires (SMWs)

SMWs were prepared using a sol-gel process. TEOS and ethyl alcohol were mixed at a molar ratio of 1:1 for 1 h and hydrochloric acid was added as a catalyst to the solutions. The AAO template was infiltrated for 8 h in the solution for the silica sol impregnation of AAO templates. After the sol-gel process, the AAO-template-filled synthesized SMWs were dried via the evaporation of the residual solvent for 2 h. After drying, the AAO template was dissolved and removed in a 3 M NaOH solution to collect the macrowires. The SMWs were purified using centrifugation at 15,000 rpm for 10 min. The collected SMWs were rinsed in ethyl alcohol and then dried in an oven.

### 2.3. Synthesis of Folic Acid-Conjugated SMWs (FA-SMWs)

After dispersing 100 mg SMWs in 4 mL ethyl alcohol, 1.2 mmol ATPES was subjected to nitrogen purging for 12 h. APTES can easily modify the surface of SMWs with primary amine groups (-NH_2_). The collected product was rinsed in ethyl alcohol and then dried in oven. After dispersion of the primary amine modified SMWs in 100 mL DMF, they reacted with EDC 1.5 eq, NHS 1.5 eq, TEA 2.0 eq, FA 1.5 eq for 12 h, resulting in a carbodiimide reaction. The FA conjugated SMWs were separated by centrifugation at 15,000 rpm for 10 min.

### 2.4. Zr-89 Labeling and In Vitro Stability Test

One mg of SMWs were placed in 5 mL vials and were dispersed in 100 µL of HEPES. Then, 37 MBq of ^89^Zr(ox)_2_ was added to the vial and allowed to react at 95 °C for 1 h. After the reaction, the product and the solvent were separated by centrifugation, and free ^89^Zr was removed by washing with 50 mM DTPA solution, and then washed two times with PBS. Centrifugation was carried out at 10,000 rpm for 5 min. FA-SMWs and Dox-FA-SMWs were also labeled in the same process. The ^89^Zr-labeled SMWs were dispersed in 100 µL of PBS for stability testing. 100 µL of 89Zr-labeled SMWs (37 MBq/mL) was added to 1 mL of HS and FBS. Tubes were stirred for seven days at 37 °C. At the selected time points (1, 2, 4, and 8 h and 1, 3, and 7 days), the in vitro stability of the ^89^Zr-labeled SMWs was evaluated using a Radio-TLC scanner. DTPA (50 mM) was used as the mobile phase and i-TLC-SG (1.0 × 10 cm^2^) was used as the stationary phase.

### 2.5. Cell Internalization

Twenty-four hours before evaluation, murine colorectal cancer cell, CT-26, and human breast cancer cell, MDA-MB-231, (both FR positive), were incubated in 24-well plates (37 °C, 5% CO_2_). After 24 h, ^89^Zr-SMW and ^89^Zr-FA-SMW (0.18 MBq/400 μL), dispersed in DMEM containing 10% FBS, were added and then incubated for 1, 4, 6, 24 and 48 h in an incubator (37 °C, 5% CO_2_). The supernatant was collected separately in tubes at each time point. Cells were carefully washed 3 times with cold PBS. Thereafter, 0.1 M sodium citrate was added and left for 5 min to remove non-internalized particles remaining on the cell membrane. A gamma-ray counter was used to measure the number of particles in the cell (n = 3).

### 2.6. PET Studies

PET studies were conducted in accordance with the animal testing guidelines and ethics approved by Dongguk University (IACUC-2021-01). PET images were measured to assess biodistribution and in vivo behavior of ^89^Zr-SMWs. For modeling tumor-bearing mice, CT-26 cells (5 × 10^6^ cells) were dispersed in 100 µL of saline and injected subcutaneously into the thighs of Balb/c mice. After allowing the injected cells to stably grow into 120 mm diameter tumors, each mouse was injected intravenously with ^89^Zr-SMWs (3.7 MBq/100 µL per mouse). The mice were under respiratory anesthesia using isoflurane and evaluated using PET for 6 days. ^89^Zr-FA-SMWs were evaluated via the same process as ^89^Zr-FA-SMWs and were injected into the tumor site to confirm the in vivo behavior of the drug carrier.

## 3. Results and Discussion

### 3.1. Characterization

#### 3.1.1. Fourier Transform–Infrared Spectroscopy (FT–IR)

The SMWs synthesis and FA binding studies were analyzed using FT–IR. The SMWs showed a strong bond between silicon and oxygen at 1080 cm^−1^ (Figure 1a, black). As a result of the FT–IR measurement of SMWs-APTES, the C-H (2950 cm^−1^) and N-H (1650 cm^−1^) bonds, which are characteristic of APTES, were confirmed, and it was verified that APTES was bonded to the SMWs surface. As a result of comparing the FT–IR spectra of SMWs-FA (Figure 1b, red) and FA (Figure 1b, orange), the C=C bond (1580 cm^−1^) and N-O bond (1500 cm^−1^) of FA appear in SMWs-FA.

#### 3.1.2. Solid-State (^13^C) Nuclear Magnetic Resonance (ssNMR)

The FA-SMWs were measured using ssNMR to confirm the conjugation of the SMW surface with FA. The results showed the aliphatic hydrocarbon of FA (10–57 ppm), aromatic hydrocarbon (114.02–149.08 ppm), aromatic carbonyl carbon (155.29 ppm), amide carbon (164.23 ppm), and carboxyl carbon (177.68 ppm), as shown in Figure 2. Based on the results of ssNMR and FTIR, we confirmed that FA was successfully conjugated on the surface of the SMWs.

#### 3.1.3. Electron Microscopy and Dynamic Light Scattering (DLS) Analysis

The AAO template was used to synthesize the uniform form of silica microwires in this study. The SMWs obtained by removing the template after synthesis were uniformly shaped, as shown in the SEM results depicted in Figure 3a. The SMWs were about 2.33 μm in length and about 200 nm in diameter, as shown in the image of Figure 3a,b. In addition, the analysis using TEM (Figure 3c,d) showed that SMWs were synthesized in wire form and that the distribution of even silicon dioxides was generally confirmed. To confirm the element distribution of SMWs, element mass ratio and mapping analysis were performed using the energy dispersive spectrum (EDS). The elemental distribution of SMWs was confirmed to be mainly composed of 49% oxygen and 24% silicon, and 10% carbon was detected in the fixation reagent used to fix the sample (Figure 4). In Figure 5, DLS results of SMWs dispersed in distilled water confirmed that they have two (453 and 2665 nm) hydrodynamic sizes, which verifies that the synthesized SMWs exist in the form of wires in the dispersion.

### 3.2. Radio Labeling and Biological studies

#### 3.2.1. ^89^Zr Labeling

In this reaction, when labeling ^89^Zr, a method of directly coordinating ^89^Zr to hydroxyl groups (-OH) on the silica surface without a chelator was used. This method is possible because Zr is an oxygen-affinity metal [31]. After the ^89^Zr labeling reaction, the labeling yield of ^89^Zr-SMWs was evaluated using a radio-TLC scanner. As shown in Figure 6, the labeling yield of ^89^Zr-SMWs was 40.96% (R_f_ = 0.15). This is low when compared with the labeling method using a chelator. However, in this study, to prevent the introduction of compounds other than FA on the surface of our drug carrier, the chelator-free labeling method was chosen, and sufficient efficiency was achieved for the biological evaluation to proceed.

#### 3.2.2. In Vitro Stability Test

For seven days after the ^89^Zr-SMWs were labeled, the in vitro stability was evaluated for HS and FBS, and measurements were made using a gamma counter. As shown in Figure 7, the ^89^Zr-SMWs showed a high stability of more than 95% for 7 days in human serum. This means that ^89^Zr was firmly positioned in the silanol group on the surface of SMWs, and is thus expected to be suitable for in vivo behavior studies of drug carriers using ^89^Zr-SMWs. The stability test conducted on FBS showed lower values than HS, but still showed around 95% stability over 7 days, demonstrating meaningful in vitro stability.

#### 3.2.3. Cell Internalization Assay

Cell internalization of SMWs was evaluated depending on the time taken to confirm cell uptake. Two samples (^89^Zr-SMWs and ^89^Zr-FA-SMWs) were prepared for the comparison of uptake according to FA binding. Cancer cell lines were evaluated using CT-26 and MDA-MB-231, and both are FR-positive cancer cell lines. As shown in Figure 8, all subjects show a rapidly high uptake, and among them, ^89^Zr-SMWs without FA, were found to be particularly high in both cancer cell lines. In addition, ^89^Zr-SMWs showed sustained internalization for 48 h and over 17.08% (green line) was measured on the MDA-MB-231, while the uptake of ^89^Zr-FA-SMW decreased after 24 h. In this way, in the evaluation of cell internalization, the fact that the FA-conjugated sample showed low levels in both cancer cell lines is not consistent with the PET study results. These results can be interpreted as the dependence of cell internalization on particle size and surface chemistry. In the case of particles with a hydrodynamic size of about 2.66 μm, such as SMWs, internalization by macropinocytosis is considered as the main mechanism and receptor-mediated endocytosis (RME) is the most active in particles with a diameter of from 100 to 200 nm. Although the mechanism of SMWs endocytosis should be studied in the future, the RME of ^89^Zr-FA-SMW is considered inefficient [32].

#### 3.2.4. PET Imaging

PET was carried out to evaluate the in vivo behavior and drug delivery effects of our drug carrier. To compare the tumor affinity of FA in ^89^Zr-FA-SMWs, ^89^Zr-SMWs were evaluated via the same process, and ^89^Zr-SMWs were also injected into the tumor site to analyze the in vivo behavior of ^89^Zr-SMWs located on the tumor. As shown in Figure 9a, 15 min after the tail vein injection, SMWs showed a high accumulation in the liver and spleen, regardless of the presence of FA. They accumulated in the tumor site from 6 h and the highest accumulation was observed after 3 days. The ^89^Zr-FA-SMWs, conjugated with FA, showed a high accumulation in the tumor site compared to ^89^Zr-SMWs and were confirmed to last longer. As can be clearly seen from the maximum intensity projection (MIP) image (Figure 9c), FA is positively involved in the tumor accumulation of drug carriers and the persistence of diagnostic signals. High accumulation in the liver and spleen is typically accumulated by reticuloendothelial system (RES) and unfortunately [8], it was observed to last for 6 days. This suggests that the surface improvement in ^89^Zr-FA-SMWs by means of biocompatibility polymers is necessary to improve drug delivery effects. ^89^Zr is reported to show a high bone affinity [33], and signals from bones and joints observed 2 days later were ^89^Zr-deviated from the drug carrier. In the case of ^89^Zr-FA-SMWs injected on the tumor site (Figure 9b), they were not only able to accumulate in the tumor, but also in the liver and spleen sequentially as they circulated through the blood, and depending on time, it was observed that the accumulation amount in the tumor decreased, and signals increased in the liver and spleen. Region of interest (ROI) results from quantitative analysis of signal intensities of tumor and major organs support PET studies. In Figure 10a,b, ^89^Zr-FA-SMWs showed a lower accumulation in the liver and spleen, particularly in the spleen, and that the accumulation increases for about a day after injection. However, in the tumor, it was confirmed that FA-SMWs accumulated more in the tumor from the initial time (Figure 10c). As a result of the tumor site injection in Figure 10d, it was visibly released out of the tumor from 1 day after injection, and it was quantitatively confirmed that the accumulation in the liver and spleen continued to increase. This shows that drug carriers designed to exhibit less affinity with tumors are disadvantageous for long-term blood circulation and can create difficulties in effective drug delivery, eventually causing clearance by RES even after reaching the tumor.

## 4. Conclusions

In this study, we synthesized micro-silica wires as drug carriers and conjugated FA to the surface to improve their tumor affinity. ^89^Zr, a radionuclide for PET, was labeled to evaluate its in vivo behavior and drug delivery effects. Through a PET study using ^89^Zr-FA-SMWs, it was possible to confirm the in vivo behavior of wire-type microparticles and to confirm their possible use as a drug carrier. The significance of their behavior after the accumulation of tumors was presented by confirming that microwire injected into the tumor site accumulated in the liver and spleen through blood circulation. In the evaluation of cell internalization, it was confirmed that the micro-scale sample to which FA was bound was not effective in cell internalization. However, apart from the evaluation of cellular internalization, ^89^Zr-FA-SMWs remained in tumors for a longer period in PET studies. Therefore, micro-scale silica wire can also be considered as a drug carrier targeting FR.

## Figures and Tables

**Figure 1 materials-14-03226-f001:**
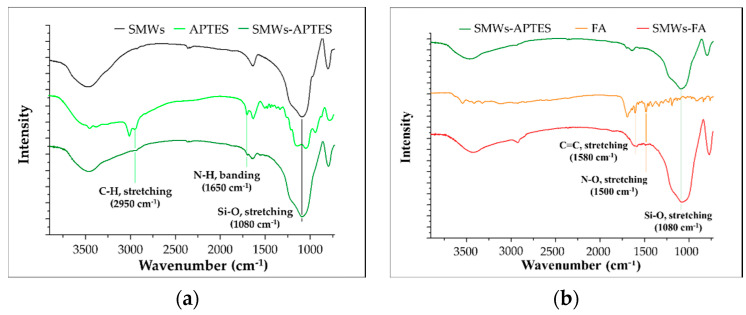
(**a**) FT–IR spectrum of APETS conjugated silica microwires (SMWs); (**b**) folic acid (orange) and folic acid conjugated SMWs (red). All samples were pelleted using KBr, and the mass ratio of the sample and KBr was 1:100.

**Figure 2 materials-14-03226-f002:**
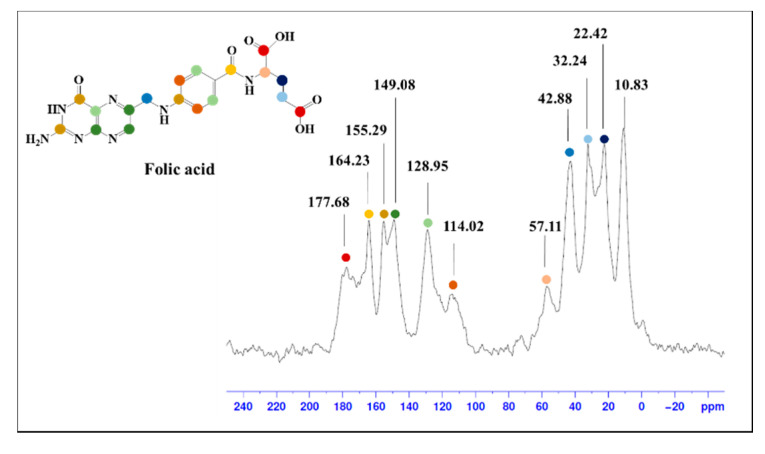
Solid-state (^13^C) nuclear magnetic resonance (ssNMR) spectrum of folic acid-conjugated SMWs. The peak in folic acid, which can be identified through ssNMR, is indicated by the same color in the molecular structure.

**Figure 3 materials-14-03226-f003:**
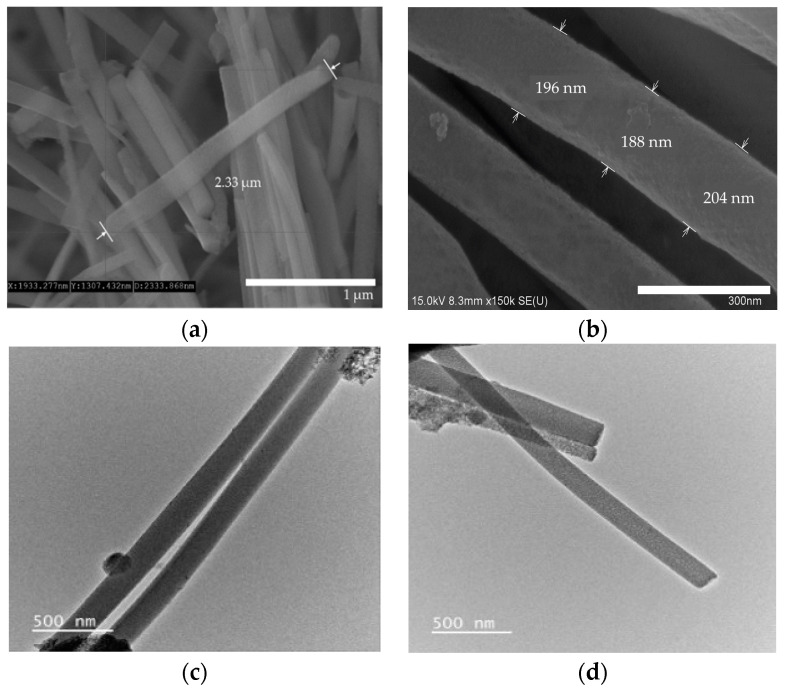
Scanning electron microscopy (SEM) images (**a**,**b**) and transmission electron microscopy (TEM) images (**c**,**d**) of SMWs.

**Figure 4 materials-14-03226-f004:**
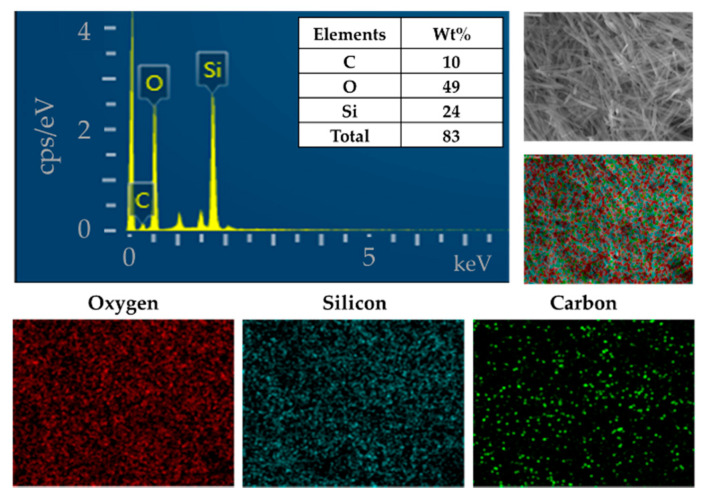
Energy dispersive spectrum (EDS) and elemental mapping images of SMWs.

**Figure 5 materials-14-03226-f005:**
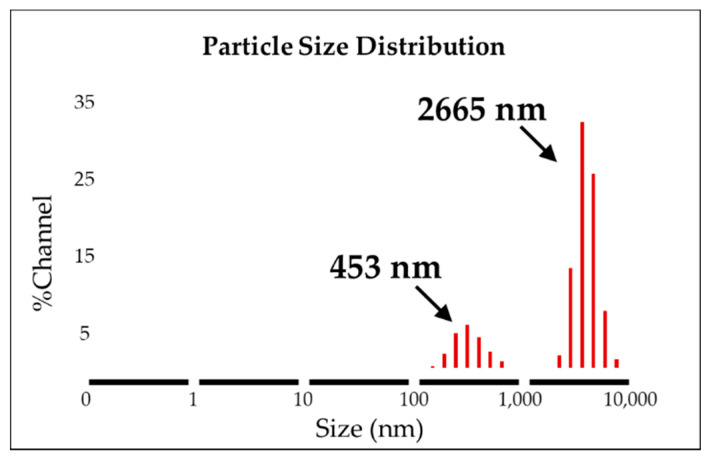
Dynamic light scattering analysis of SMWs (solvent: distilled water).

**Figure 6 materials-14-03226-f006:**
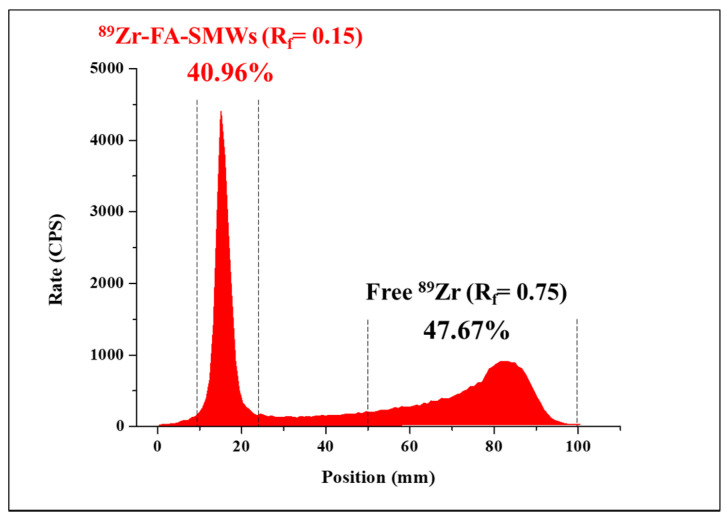
The labeling yield of ^89^Zr-FA-SMWs was confirmed using radio-TLC. ^89^Zr labeled FA-SMWs (40.96%) and free 89Zr (47.67%) were displayed using i-TLC-SG (1.0 × 10 cm^2^) as the solid phase and the 50 mM of DTPA as the mobile phase.

**Figure 7 materials-14-03226-f007:**
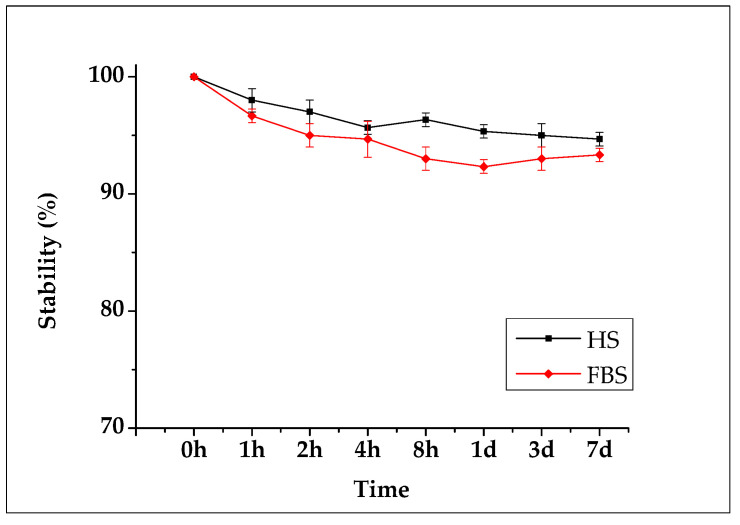
Biological stability evaluation conducted using human serum (black line) and fetal bovine serum (red line). Measured for 7 days using an automatic gamma counter (n = 3).

**Figure 8 materials-14-03226-f008:**
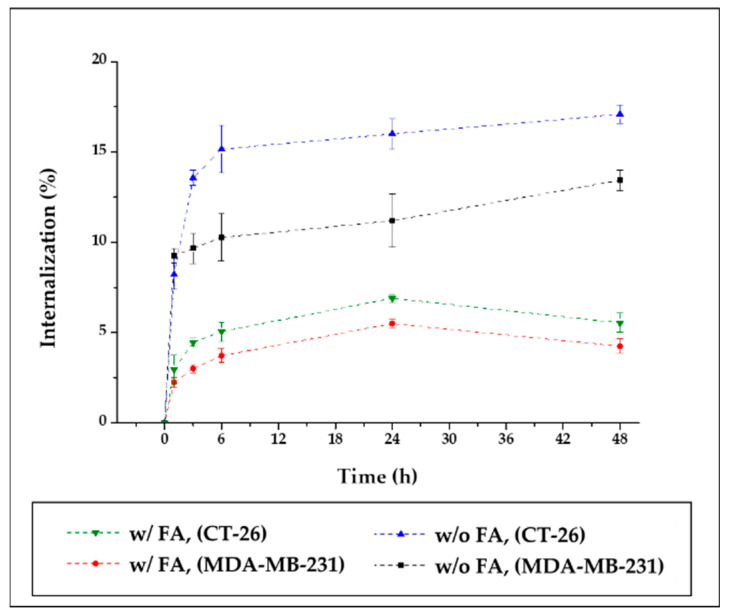
Cell internalization assay of ^89^Zr-Fa-SMWs and ^89^Zr-SMWs. Evaluation using folate receptor positive CT-26 (murine colorectal cancer cell) and MDA-MB-231 (human breast cancer cell) (n = 3).

**Figure 9 materials-14-03226-f009:**
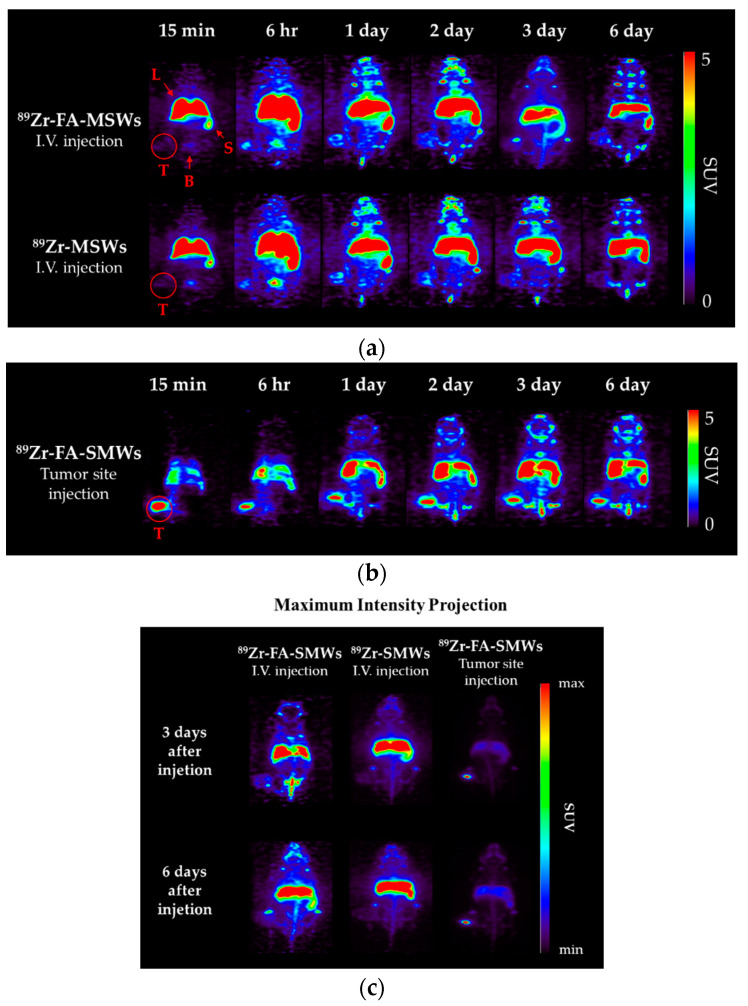
Positron emission tomography (PET) images of CT-26-bearing female Balb/c mice (5 weeks). Mice were scanned using a 5 min static mode at time points after (**a**) intravenous injection of ^89^Zr-FA-SMWs and ^89^Zr-SMWs, (**b**) tumor site injection of ^89^Zr-FA-SMWs and (**c**) maximum intensity projection (MIP) images.

**Figure 10 materials-14-03226-f010:**
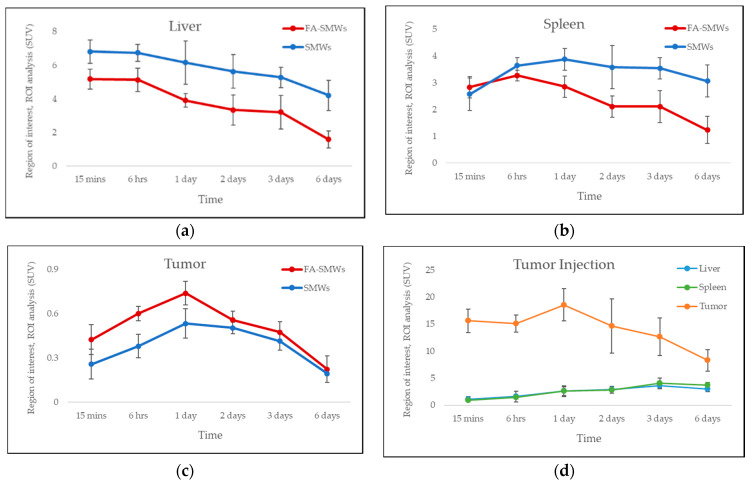
Region of interest (ROI) analysis of PET studies. Standard uptake value of (**a**) liver, (**b**) spleen, and (**c**) tumor for ^89^Zr-FA-SMWs (FA-SMWs, red) and ^89^Zr-SMWs (SMWs, blue). Standard uptake value (**d**) tumor site injection of ^89^Zr-FA-SMWs (n = 3).

## Data Availability

The data presented in this study are available on request from the corresponding author.

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
