# Peer review of "Synthesis of Zr-89-Labeled Folic Acid-Conjugated Silica (SiO_2_) Microwire as a Tumor Diagnostics Carrier for Positron Emission Tomography"

_materials, 2021, doi:10.3390/ma14123226_

Round 1

Reviewer 1 Report

In this manuscript by Kim et al, a Zr-89 labeled folic acid conjugated SiO2 microwire was synthesized. This material showed low receptor-mediated uptake in cell experiments, but allowed long-term accumulation in tumor regions in vivo. This paper should be accepted after the following issues are carefully addressed:

  1. In section 3.1.3, the authors claimed that the diameter of the SMWs was about 200 nm. A more detailed statistical analysis of the diameters and lengths should be provided to show the monodispersity of the SMWs.
  2. The authors could strengthen their claim on the distribution of silicon oxide by showing some elemental mapping results along with the TEM images.
  3. In section 3.2.3, the following experiments are suggested to strengthen the results: (1) Folate receptor-negative cell lines should be tested as a negative control. (2) The internalization experiments should also be performed at 4 degrees to preclude the non-specific pinocytosis.
  4. As shown in section 3.2.3, it seems that FR-mediated internalization is disadvantageous in this case, which seems to be counterintuitive. The authors should do a literature search, and see if similar result has been reported previously.
  5. In section 3.2.4, a quantitative study of biodistribution of the materials should be provided. In particular, the signal in the tumor, liver, spleen and surrounding normal tissues should be analyzed.

Reviewer 2 Report

The manuscript “Synthesis of Zr-89-Labeled Folic Acid-Conjugated Silica (SiO2) Microwire as a Tumor Diagnostics Carrier for Positron Emission Tomography” synthesized a homogeneous silica microwire for drug delivery, and evaluated its behavior in vivo and accumulation, then confirmed that the folic acid-conjugated sample remains at the tumor site for a relatively longer time. This manuscript needs minor revision before considering published at Materials.

  1. In the abstract, there is no abbreviation for folic acid (FA). FA is used directly below. It is suggested to add.
  2. The introduction is an important part of a paper. The introduction written by the author introduces a lot of basic content, but does not tell readers why it is necessary to carry out the following research. For example, the shape of the drug carrier described in the third paragraph will affect the effect, but it does not explain what problems exist and why the study of silica microfilaments is needed. The third paragraph says “Along with size, particle shapes can affect…”. However, the effect of size has not been described. “…because the researcher adopted an 'upward' method of production.”. A brief introduction to the “upward” method of production is recommended. And this paragraph has no references. For porous microwire, the author may refer to a new reference: Green Energy & Environment, 2020, 5(3): 303-321. To sum up, the introduction will make readers confused.
  3. In 2.1 Materials, only HEPES has marked the concentration, and the other reagents have not marked the concentration. It is suggested to unify the format.
  4. In the part of result and discussion, the author only described the content in the figure. I think the author should add some explanations about the mechanism and make a theoretical discussion, which will make the article more detailed and reliable. For example, how does FA affect the experimental results and why does it produce different effects. It is better to add the contents of pharmacological action and pharmacokinetics.

Round 2

Reviewer 1 Report

The manuscript has been strengthened after revision. I would like to see the acceptance of the manuscript in its current form.